# Additively Manufactured Flexible EGaIn Sensor for Dynamic Detection and Sensing on Ultra-Curved Surfaces

**DOI:** 10.3390/s25010037

**Published:** 2024-12-25

**Authors:** Jiangnan Yan, Jianing Ding, Yang Cao, Hongyu Yi, Limeng Zhan, Yifan Gao, Kongyu Ge, Hongjun Ji, Mingyu Li, Huanhuan Feng

**Affiliations:** 1Sauvage Laboratory for Smart Materials, Shenzhen Key Laboratory of Flexible Printed Electronics Technology, Harbin Institute of Technology, Shenzhen 518055, China; 2State Key Laboratory of Advanced Welding and Joining (Shenzhen), Harbin Institute of Technology, Shenzhen 518055, China

**Keywords:** precision additive manufacturing, modular sensing unit, electronic skin, ultra-curved surface

## Abstract

Electronic skin is widely employed in multiple applications such as health monitoring, robot tactile perception, and bionic prosthetics. In this study, we fabricated millimeter-scale electronic skin featuring compact sensing units using the Boston Micro Fabrication S130 (a high-precision additive manufacturing device) and the template removal method. We used a gallium-based liquid metal and achieved an inner channel diameter of 0.1 mm. The size of the sensing unit was 3 × 3 mm^2^. This unit exhibited a wide linear sensing range (10–22,000 Pa) and high-pressure resolution (10 Pa) even on an ultra-curved surface (radius of curvature was 6 mm). Sliding was successfully detected at speeds of 8–54 mm/s. An artificial nose with nine sensing units was fabricated, and it exhibited excellent multitouch and sliding trajectory recognition capabilities. This confirmed that the electronic skin functioned normally, even on an ultra-curved surface.

## 1. Introduction

Electronic skin shows remarkable properties; for example, it achieves high conformity when affixed to nonuniform surfaces [1,2,3,4,5,6,7,8,9,10] and emulates the flexible properties of the human epidermis to facilitate supple tactile discernment [11,12,13,14]. It is widely used in health monitoring, human–robot interactions, and bionics [15,16,17,18]. Eutectic gallium-indium (EGaIn) is considered the most promising material for fabricating electronic skin. Zhou et al. fabricated a flexible electronic device using polyvinyl alcohol-encapsulated liquid gallium [19]. The device exhibited good durability and stability when it was bent and twisted, owing to the flexibility of the substrate and liquid metal. Kramer et al. used photolithography to fabricate a graphic master. They employed stacking and layer-by-layer bonding through reverse molding to obtain a flexible polydimethylsiloxane (PDMS) substrate with double-layer serpentine channels. They infused these channels with EGaIn to obtain a wearable tactile keyboard [20]. The electronic skin utilized a quasiplanar intersection network constructed using double-layer serpentine circuits to sense the position, intensity, and time of applied pressure based on the electrical signal response at an intersection. However, an external pressure of at least 100 kPa was required to cause a discernible change of approximately 5% in the output voltage. Yang et al. mixed EGaIn with Ecoflex using the sacrificial particle method to obtain a foamy composite elastomer [21]. Subsequently, liquid gallium was sprayed to prepare electrodes on both sides of the composite elastomer, which served as the dielectric layer [22,23,24,25,26,27,28,29]. The resulting flexible capacitive sensor array accurately identified the position of touch points [30,31,32,33,34,35,36,37]. The measurement precision decreased during bending. This was primarily attributed to the large size of the sensing unit, which experienced substantial perturbations under deformation. A reduction in the size of the sensing unit mitigated the impact of bending, thereby highlighting the importance of miniaturization.

The challenges in the miniaturization of sensing units include difficulties in EGaIn injection and complex sensor fabrication [38,39,40,41]. Typically, these issues are resolved by utilizing vacuum degassing and printed circuitry [42,43,44,45,46]. To address the requirement of a gas outlet when perfusing liquid metals, Dickey et al. used a vacuum chamber to remove gas from a microchannel and elastic substrate and showed that it was as small as several microns [47]. The filling of microns and branched dead-end structures is straightforward and requires minimal human intervention. Xi et al. reduced the line size to 0.1 mm and fabricated ultrathin micro-tubes with a diameter of approximately 0.12 mm in PDMS by employing thin copper wires. They injected EGaIn inside the tubes to obtain an ultrathin microtube sensor [48]. The sensor exhibited a minimum resolution of 5 mN and a high sensitivity of 68 N. The diameter of the sensor, which was comparable to that of human hair, allowed for the almost imperceptible monitoring of the human pulse. Zhang et al. printed liquid–metal lines on a prestrained PDMS substrate and subsequently released the strain, thereby decreasing the width and spacing of the metal lines [49]. The UV–ozone surface modification of a PDMS substrate prior to printing can improve the adhesion between a liquid metal and the substrate. This method can be utilized to reduce the resolution between metal lines to 0.03 mm. Park et al. proposed a high-resolution printing method for liquid EGaIn. This method was used to obtain patterns with a minimum linewidth of 1.9 mm through direct printing, which allowed for the preparation of stretchable electrodes with fine structures [50]. The relatively strong oxide skin maintained the printed 3D structure after printing. We presented the results for centimeter-level electronic skin in our previous work [36], where the resistance drift was significant in environments with small curvature radii. Despite the implementation of diverse methodologies and advancements, mass production of liquid metal flexible sensors remains a significant challenge. The complex fabrication processes have severely hindered their practical and widespread application. Therefore, a new manufacturing method must be developed, and the associated challenges must be resolved.

In order to mitigate signal shifts on highly curved surfaces, we fabricated electronic skin using the template removal method. This method is both simple and cost-effective, offering a feasible approach for the mass production of EGaIn sensor arrays. The size of a sensor unit was 3 × 3 mm^2^, and its linear sensing range was 10–22000 Pa with a pressure resolution of 10 Pa on an ultra-curved surface (radius of curvature was 6 mm). The spatial resolution was 8 mm, which was comparable to that of the human nose [33]. A critical spacing of 2 mm was used to prepare a 3 × 3 mm^2^ sensing array that could accurately sense multitouch and sliding tracks on an ultra-curved surface while detecting motion speeds of 8–54 mm/s. Our sensor unit is sufficiently flexible to be applied to ultra-curved surfaces while providing excellent sensing performance. Furthermore, it has high potential for applications in health monitoring and robotics, thereby advancing the field of tactile sensing technology and addressing key challenges in curved surface sensing.

## 2. Materials and Methods

### 2.1. Materials

Materials: EGaln (Shuochen, Changge, China), PDMS (Dow Corning, Midland, MI, USA), and Ecoflex-0030 (Smooth-On, Macungie, PA, USA) are utilized as the conductive functional material and packaging structural materials, respectively. NaOH (Damao, Tianjin, China) is used for the oxidation of EGaIn. Copper wires and weights for testing are purchased from the market.

### 2.2. Modular Flexible Electronic Skin Fabrication

We used EGaIn as a conductive functional material to prepare modular electronic skin using the template removal method. First, a predesigned small-scale circuit pattern and mold base plate were printed as resin templates using S130 (Boston Micro Fabrication, Shenzhen, China). The size of the sensing area was 3 × 3 mm^2^, and the circuit density was 3.2 mm^−1^, and the circuit width was 0.1 mm. The thickness of the sensing layer was 1 mm, ensuring good mechanical strength while keeping the sensor as thin as possible. Subsequently, the sample was coated with Ecoflex, heated, and cured at 45 °C. The cured Ecoflex was peeled off such that the circuit pattern in the resin template was successfully transferred to the bottom surface of the flexible Ecoflex substrate. Then, a PDMS coating was applied to a blank mold and heated at 45 °C for an appropriate duration to obtain the initial shape, resulting in the formation of a semi-cured PDMS film. The thickness of the PDMS substrate was 1 mm. The peeled flexible Ecoflex substrate and semi-cured PDMS film were further heated and cured at 45 °C until they were completely bonded. Subsequently, the original EGaIn was mixed with a 0.2 mol·L^−1^ NaOH solution (24 g of EGaIn and 20 mL of NaOH solution). After undergoing ultrasonic cleaning for 10 min to remove oxides, the mixture was stirred with a magnetic stirrer at 200 r/min for 4 h to induce oxidation. Then, The oxidized and modified EGaIn was injected into the circuit channel using a small syringe and then guided out through ports on both sides using copper Cu wires. The copper (Cu) wire used in our sensor had a diameter of 0.035 mm, was inserted into the channel to a depth of approximately 1 mm, and then sealed with epoxy resin AB glue (the A and B parts of the glue were pre-mixed in 1:1 portion for 5 min, and then dried at 30 °C for 5 h), resulting in modular electronic skin with a size of 3 × 3 mm^2^.

### 2.3. Unit Properties Test

In this experiment, weights were used to apply loads to millimeter-scale sensing units, and a multimeter (DMM7510, Keithley, Beaverton, OR, USA) was utilized to measure the resistance responses of these units under various loading conditions. To ensure uniform force distribution, thin pads of 3 × 3 mm^2^ were pre-placed on the sensing units during loading. To measure the linear sensing range and response time of the sensing units, weights of 0.005 g (5 Pa), 0.01 g (10 Pa), 0.05 g (55 Pa), 0.2 g (220 Pa), 1 g (1100 Pa), 5 g (5500 Pa), 20 g (22,000 Pa), and 50 g (55,000 Pa) were sequentially applied and removed from the sensing units, and their responses were detected. To assess the pressure resolution of the sensing units, weights of 10 g, 5 g, 1 g, 0.2 g, 0.05 g, 0.01 g, and 0.005 g were incrementally added to the sensing units at a time interval of 100 s until no significant change in the resistance signal of the sensing unit was observed at the time of each incremental addition. To evaluate the frequency detection performance of the measurement sensor unit, the function generator is adjusted to output sinusoidal waveforms at frequencies of 1 Hz, 2 Hz, 10 Hz, 25 Hz, 50 Hz, and 100 Hz, respectively. A sinusoidal deformation load is applied to the surface of the sensor unit via the front end of the exciter. The response of the sensor unit is then captured using a multimeter.

### 2.4. Static Touch Test

To evaluate the multi-touch capability of millimeter-scale electronic skin, a multichannel module of a multimeter was utilized to simultaneously monitor the resistance signals from nine sensing units. These units were loaded with weights of 2 g, 3 g, 5 g, 8 g, 10 g, 12 g, 15 g, 18 g, and 20 g, respectively, in order to obtain the corresponding resistance signal responses from each sensing unit.

### 2.5. Dynamic Sliding Test

A weight of 19.17 g is used to apply pressure, and continuous smooth shims allow the weight to slide with little friction. When testing the relation curve between load and speed, the weight is pulled by the stepping motor to move at a uniform speed. In the subsequent track tests, the nonlinear sliding is realized manually.

### 2.6. Curved Surface Test and Nose Surface Bionic Model Test

To investigate the impact of various curved surface conditions on the sensing performance of millimeter-scale sensing units, the BMF P150 3D printer was utilized to print thin resin pads with curvature radii of 10 mm, 8 mm, 6 mm, and a flat surface to achieve bending of the sensing units. Under these four curvature conditions, weights of 10 g, 1 g, 0.2 g, 0.05 g, 0.01 g, and 0.005 g were progressively added to the pads at intervals of 60 s to obtain continuous loading resistance response curves and analyze their pressure resolution. Subsequently, under the same four curvature conditions, weights of 0.01 g, 0.05 g, 0.2 g, 1 g, 5 g and 20 g were loaded and unloaded, respectively, to capture the resistance response information during loading.

To prepare a bionic nasal tip model, the initial step involves printing a cavity with the geometric shape of a nasal tip on its inner wall using a fused deposition modeling 3D printer X400 (Huitianwei, Beijing, China). Subsequently, nine sensing units, each with a size of 3 × 3 mm^2^, are attached to appropriate positions on the inner wall of the cavity in an array formation, spaced 2 mm apart, using double-sided adhesive tape. The entire cavity is then filled with uncured Ecoflex, which is left to cure completely at room temperature for 12 h. After the curing process, the cavity is demolded to obtain the bionic model. When validating the multi-touch and sliding sensing performance, a multichannel module of a multimeter is utilized to monitor the response of the sensing units’ signals.

## 3. Results and Discussion

Modular electronic skin was fabricated through the template removal method. Initially, resin templates with preset circuit patterns and molds were printed using BMF S130 (Figure 1A). These templates were then coated with Ecoflex, heated, and cured at 45 °C (Figure 1B). Upon stripping the cured Ecoflex, the circuit patterns were successfully transferred to the flexible Ecoflex substrate’s bottom (Figure 1C). Next, a layer of PDMS was coated onto a blank mold and partially cured at 45 °C to create a semi-cured film (Figure 1D). This semi-cured PDMS film was then bonded to the flexible Ecoflex substrate through further curing at 45 °C (Figure 1E). In our previous work, we utilized pure EGaIn to fabricate centimeter-scale electronic skin [36]. However, when attempting to further scale down to create millimeter-scale electronic skin, pure EGaIn tends to cause injection difficulties, susceptibility to open circuits, and poor bending performance due to its high surface tension and inadequate wettability. Therefore, we tackled this challenge by utilizing controlled oxidative modification of EGaIn to reduce its surface tension. The modification method and subsequent analysis of the results are presented in Appendix A. Specifically, it involves mixing EGaIn with a NaOH solution and stirring with a magnetic stirrer to oxidize the surface layer, forming gallium oxide, which is then fragmented and dispersed into the interior of EGaIn. Stirring for 4 h completes the modification process (Figure 1F). The oxidized EGaIn was injected into the circuit channels using a small syringe (Figure 1G), and copper wires were connected through ports on both sides (Figure 1H), ultimately resulting in the formation of the modular electronic skin. As a result, we reduced the size of the sensing units from 1 cm^2^ to 3 × 3 mm^2^, expanded the linear sensing range from 100–20,000 Pa to 10–22,000 Pa, and improved the pressure resolution from 20 Pa to 10 Pa. Additionally, the electronic skin developed in this work exhibited a 20-fold increase in sensing accuracy on ultra-curved surfaces with a curvature radius of 6 mm, improving from 200 Pa to 10 Pa. Therefore, it aids this electronic skin in achieving a high level of integration, thereby achieving accurate and sensitive recognition and the perception of static touch and dynamic sliding under external pressure loads (Figure 1I,J). The resulting electronic skin exhibited excellent touch performance (Figure 1K) and can be applied to more complex parts of the human body (Figure 1L).

Figure 2 demonstrates the pressure resolution, response time, linear sensing range, and frequency detection performance of the sensing unit. The detailed principle of the change in sensor resistance, as illustrated in Appendix A, is primarily due to the variation in the resistance of the liquid metal circuit. When pressure is applied to our sensing unit, the cross-sectional area of the microchannel decreases, resulting in a corresponding change in resistance.

We carry out a weight-loading experiment on the sensing unit. The change in resistance under different loads and the results are shown in Figure 2A. The value of R^2^ for the linear fitting result between the resistance and pressure load is 99%, indicating an excellent linear relationship. Therefore, the linear sensing range of the sensing unit is 10–22,000 Pa (0.01–20 g). The response times of the sensing unit are obtained under six pressure conditions, as shown in Figure 2B. As the pressure increases in a range of 10–22,000 Pa, the response time gradually increases from 0.05 s to 0.14 s, indicating an extremely high response speed. The pressure resolution of the sensing unit is determined to estimate the response to the pressure load through weight accumulation. Figure 2C shows that the sensing unit can accurately respond to a minimum load of 0.01 g (10 Pa), i.e., it has a high-pressure resolution.

Pressure loading in a range of 55–11,000 Pa produces a strong resistance response. The capability of the sensing unit to respond to the frequency of applied vibration is evaluated in a range of 1–50 Hz, and the results are shown in Figure 2D. The resistance response corresponding to the frequency is calculated from the resistance response curve. The calculated values are 0.99 Hz, 2.00 Hz, 10.01 Hz, 25.03 Hz, and 49.72 Hz. The response frequency and function of the sensing unit are calculated using the resistance response. The frequency of the sinusoidal waveform pre-set by the generator is constant. The sensing unit exhibits excellent performance in determining the frequency of periodic deformation loads in a range of 1–50 Hz. Figure 2E not only displays the specific performance of each sensor (details in Table 1) but also compares the performances of several similar sensors, highlighting that the EGaIn sensor unit demonstrates particularly outstanding comprehensive performance, especially in its response to sliding loads, low pressures, and frequency detection.

Sensing units are integrated into an array of flexible electronics. The critical distance between two adjacent sensing units is determined through weight-loading experiments. For the sample shown in Figure 3A, a pressure load of 22,000 Pa is applied to the left unit for 60 s, and the resistance response of the right unit is measured using a multimeter. Figure 3B,C show the resistance responses for distances of 1 mm and 2 mm between adjacent sensing units. When the pressure load is applied with a distance of 1 mm between adjacent sensing units, the continuous flexible substrate causes the deformation of the right unit by a small amount of stress, and the resistance increases by approximately 0.19 mΩ. Additionally, crosstalk occurs during the resistance response of the two sensing units. A distance of 2 mm is sufficient to ensure that the right unit is minimally affected by the pressure load on the left unit. Therefore, the critical spacing between the sensing units in the millimeter-scale electronic skin is 2 mm. The side length of each sensing unit is 3 mm. The largest distance between two adjacent units is the spatial resolution of the modular electronic skin (Figure 3A), which is 8 mm. The electronic skin exceeds the spatial resolution of the human skin on the nose [33].

As shown in Figure 3D–I, the electronic skin consists of nine sensor units and accurately responds to planar static multitouch. The response results are compared with those obtained using actual loads by simulating two-point, three-point, and nine-point touches using nine weights in a range of 2–20 g (Figure 3D–F), and the results are shown in Figure 3G,H. The sensing units with touch respond effectively, whereas those without touch remain unresponsive. This further verifies the critical distance determined in our previous experiment. Additionally, the crosstalk between the sensing units is successfully eliminated even when they are surrounded by additional sensing units. The bar height in the graph represents the touch pressure measured by the sensing units. The root-mean-square error of these measured pressure values is lower than 150 Pa, demonstrating the exceptional accuracy of the electronic skin for touch pressure sensing. Therefore, the electronic skin has an exceptional ability to perceive static multitouch.

The electronic skin can perceive the speed of moving pressure loads on the basis of the sequential response of the sensing units and the amplitude of the resistance peak. A metal ball (19.17 g) is used to determine the displacement speed along the first row of sensing units (Figure 4A). The resistance response curve is shown in Figure 4B. The rolling speed of the metal ball can be calculated using the following formula:(1)$$\bar{\nu }=2\times 5\;mm/t_{1}+t_{2}$$
where $\bar{\nu }$ is the average rolling speed, $t_{1}$ is the rolling time from unit 1 to unit 2, $t_{2}$ is the rolling time from unit 2 to unit 3.

The electronic skin effectively detects speeds in a range of 8–54 mm/s. The resistance increment (ΔR) shown in Figure 4B is used to determine the equivalent pressure load applied on the sensing unit during the rolling process, and the results are shown in Figure 4C. As expected, the static load caused by the mass of the metal ball remains constant, but the calculated equivalent pressure load decreases as the speed increases. When an object moves at a certain speed, the equivalent pressure load reflected by the resistance response of the sensing unit differs from the static load generated by the mass of the object. The correlation coefficient (R^2^) of the linear fit of the acquired data exceeds 98%. This linear relationship is used to convert the equivalent pressure load into the static load caused by the mass of the object. A prefactor is multiplied with the equivalent pressure load to calculate the static load. The prefactor is given by
(2)$$k_{2}=0.9878\cdot \bar{\nu }-0.0953$$

A soft brush tip is gradually slid across the electronic skin while a pressure load is applied to test its trajectory recognition and load response capabilities for dynamic sliding. The sliding path and response results are shown in Figure 4D. During the sliding process, the sensing units located on the path respond sequentially, presenting a preset sliding track. As the nib of the brush is used for sliding, it is challenging to obtain an accurate value of the pressure load. However, we evaluate the magnitude of the applied pressure using the response results. The pressure applied during the sliding process is approximately 2110 Pa according to Figure 4D, and 2310 Pa according to Figure 4E. The response of the electronic skin to the sliding of three nibs is shown in Figure 4F. The histogram shows the main pressure points in the sliding process as follows: 1 (units 1, 4, and 7), 2 (units 2, 5, and 8), and 3 (units 3, 6, and 9). These numbers represent the responses of the nine sensing units in the column units. The applied pressure is higher when the sliding pressure point is in the second column (approximately 5120 Pa) and lower when it is in the other columns (2578 Pa and 1332 Pa, respectively). According to the above results, the electronic skin exhibits good perception ability for dynamic sliding. The sliding trajectories of single-point and multipoint loads are accurately identified. Additionally, the displacement speed is 8–54 mm/s, and the pressure value of the load is determined.

The effect of the curved surface on the sensing performance of the electronic skin is experimentally examined. We set the minimum curvature radius based on the value selected for the centimeter-scale sensor [36] because a minimum curvature radius of 6 mm is appropriate for the curved surface environment of the nose tip. A flat surface and curved surfaces with radii of 6 mm, 8 mm, and 10 mm are used to investigate the influence of surface curvature on the performance of the millimeter-scale sensing unit. The bending of the sensing unit depends on the curvature of the resin backing plate. The model and images of the backing plate and unit are shown in Figure 5A–E.

The pressure resolution of the sensing unit is determined through weight accumulation experiments, in which the sensing unit is bent to attain four predetermined curvature radii; the results are shown in Figure 5F. The sensing unit does not provide an effective resistance response to a pressure load of 5 Pa; however, it accurately senses a pressure load of 10 Pa. Therefore, the sensing unit exhibits the same high-pressure resolution (10 Pa) when subjected to bending with curvature radii of 6 mm and higher.

Figure 5F shows the four stepwise loading response curves. The resistance increases with the degree of bending. Similarly, when no pressure load is applied, the resistance of the sensing unit increases by approximately 10.73 mΩ under bending with a curvature radius of 6 mm, as compared to the planar environment. Therefore, it is necessary to add a basic resistance value (R_0_) to the linear relationship between the response of the electronic skin in a curved surface environment and the resistance of the sensing unit. The resistance responses of the sensing unit under different curved surface conditions are examined using weight loading experiments, and the results are shown in Figure 5G. Under the same pressure, the sensing unit produces a similar resistance response (ΔR/R_0_) irrespective of the bending conditions. Furthermore, the correlation coefficient (R^2^) of the linear fitting exceeds 99%, indicating a linear response range of 10–22,000 Pa.

In summary, on an ultra-curved surface such as the nose, the millimeter-scale sensing unit performs linear sensing in a range of 10–22,000 Pa with a pressure resolution of 10 Pa. Therefore, it can be used for bionic tactile sensation on the nose.

We construct a bionic model of the nose based on Ecoflex by embedding nine sensing units on the nose surface. We test the recognition and sensing abilities by conducting multitouch and sliding experiments, and the results are shown in Figure 6. A relatively low pressure (not exceeding 2500 Pa) can be distinguished from the color comparison during the three-point touch experiments, with the pressure on unit 1 being the highest (approximately 2140 Pa), the specific data are shown in Table 2. The response results of the sliding experiment clearly present the preset sliding trajectory and provide the online touch pressure during the sliding process. The online pressure during sliding is relatively high (exceeding 3000 Pa) and uniform, with specific data detailed in Table 3.The multitouch, trajectory, and pressure recognition capabilities of the electronic skin on the ultra-curved nose surface are verified, thereby demonstrating its high potential for dynamic detection and sensing.

## 4. Conclusions

We successfully fabricated a millimeter-scale electronic skin sensor based on EGaIn through additive manufacturing and template removal methods. The sensor demonstrated significant advantages in dynamic detection and perception on ultra-curved surfaces, not only bringing new breakthroughs to the field of tactile sensing technology but also providing robust support for applications in health monitoring and robotics.

Compared to previous research, our work achieved important improvements and innovations in several aspects. First, in terms of sensor miniaturization, we successfully reduced the size of the sensing units to 3 × 3 mm^2^, significantly enhancing spatial resolution and making it more suitable for complex curved environments. Second, the sensing performance was comprehensively optimized, including a wide linear sensing range (10–22,000 Pa), high-pressure resolution (10 Pa), and precise recognition of sliding trajectories and speeds. These enhancements were primarily due to the innovative application of EGaIn materials and the effective implementation of the template removal method.

Furthermore, our work validated the sensor’s capability for multitouch and sliding track recognition on ultra-curved surfaces by constructing a biomimetic nose tip model. The spatial resolution of the sensing unit (8 mm) is comparable to that of the human nose. It can detect up to nine independent touch points and identify the sliding track of single or multiple loads. Furthermore, it can sense the sliding speed in a range of 8–54 mm/s and spontaneously detect the loading pressure. Experiments show that the sensing unit can achieve good linear sensing using the change in the relative resistance (ΔR/R_0_) to describe the resistance signal response and eliminate the drift caused by the curved surface. This implies that the unit be used on ultra-curved surfaces.

Since EGaIn liquid metal solidifies below 15.7 °C, the sensor faces significant limitations in low-temperature environments. Future research will focus on developing low-temperature stable materials, optimizing sensor design, and studying low-temperature effects with compensatory strategies. Additionally, to improve the sensor’s ability to detect subtle changes in texture and fluids, we will explore incorporating ciliary structures and integrating multiple sensing technologies into a multimodal system.

## Figures and Tables

**Figure 1 sensors-25-00037-f001:**
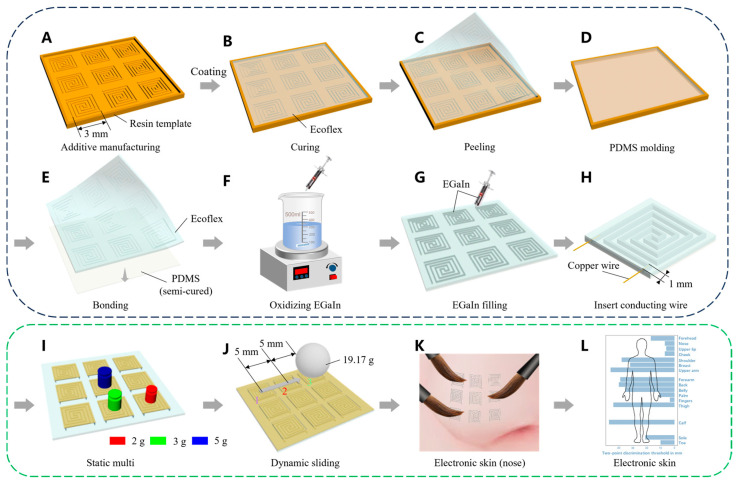
This is a figure. Schematic of preparation and performance test of EGaIn electronic skin. (**A**) Line template additive manufacturing. (**B**) Curing. (**C**) Peeling. (**D**) PDMS molding. (**E**) Ecoflex substrate is bonded to semi-cured PDMS. (**F**) Oxidizing EGaIn. (**G**) EGaIn filling. (**H**) Insert conducting wire. (**I**) Multitouch detection. (**J**) Sliding speed detection. (**K**) Multitouch test using bionic nose. (**L**) Two-point discrimination threshold and the spatial resolution of the human body.

**Figure 2 sensors-25-00037-f002:**
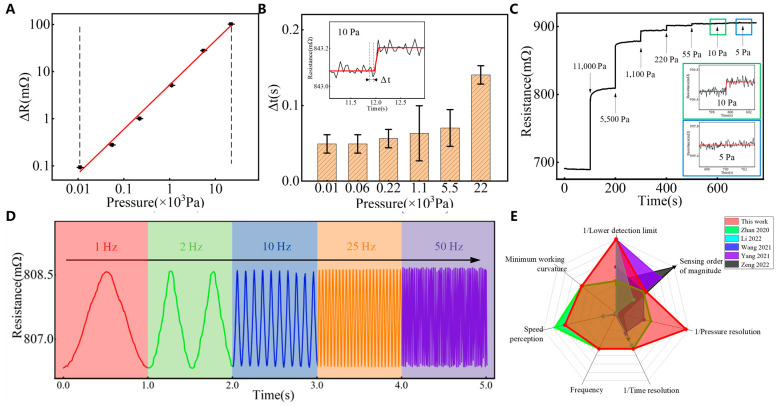
Response performance characterization of the single sensor unit. (**A**) Linear response. (**B**) Response time. (**C**) Pressure resolution. (**D**) Frequency detection. (**E**) Performance comparison. Zhan [36], Li [51], Wang [52], Yang [53], Zeng [54].

**Figure 3 sensors-25-00037-f003:**
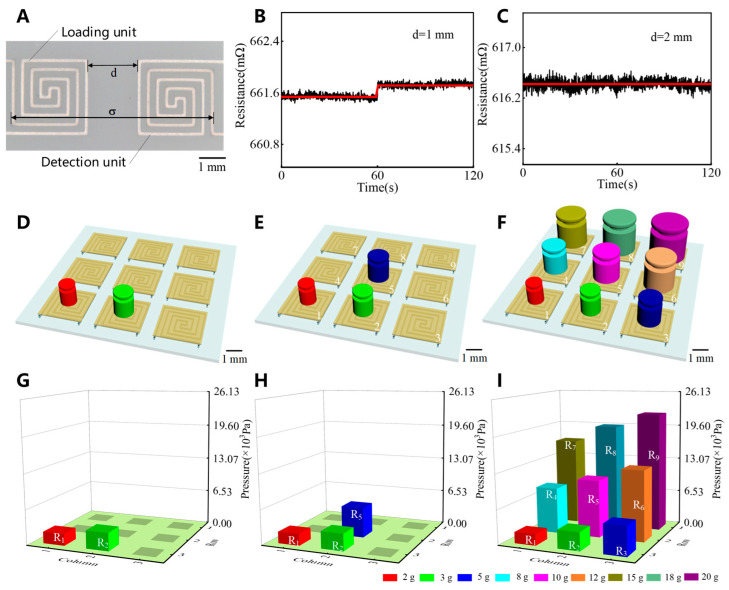
The effect of sensor array unit spacing on sensing performance and static multitouch test. (**A**) Loading unit and detection unit. (**B**) Response result (d = 1 mm). (**C**) Response result (d = 2 mm). (**D**) Two-point touch. (**E**) Three-point touch. (**F**) Nine-point touch. (**G**–**I**) Resistance response test results.

**Figure 4 sensors-25-00037-f004:**
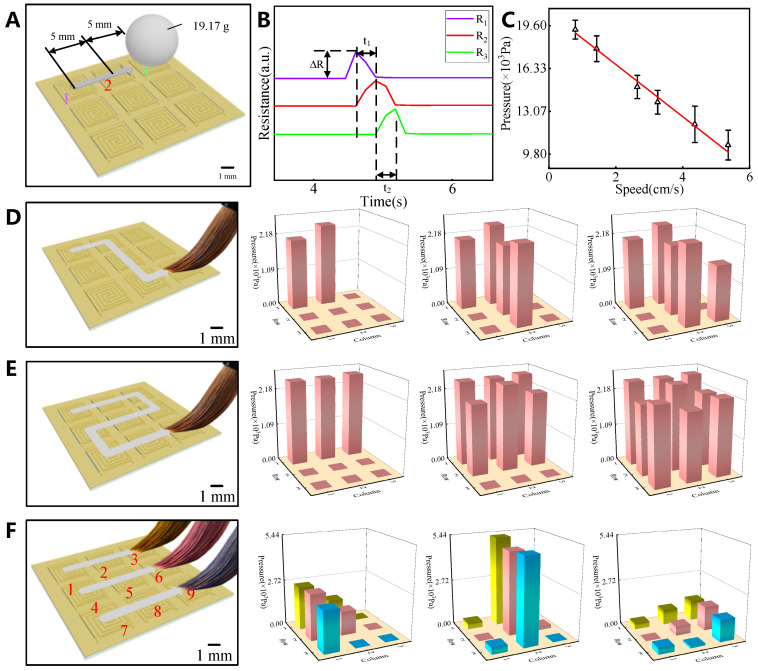
Dynamic sliding test. (**A**) Sliding speed test. (**B**) Response of resistance signal to sliding. (**C**) Relationship between sliding speed and equivalent load. (**D**) “Z”-slip and sensor array response. (**E**) “2”-slip and sensor array response. (**F**) Three-point sliding and sensor array response.

**Figure 5 sensors-25-00037-f005:**
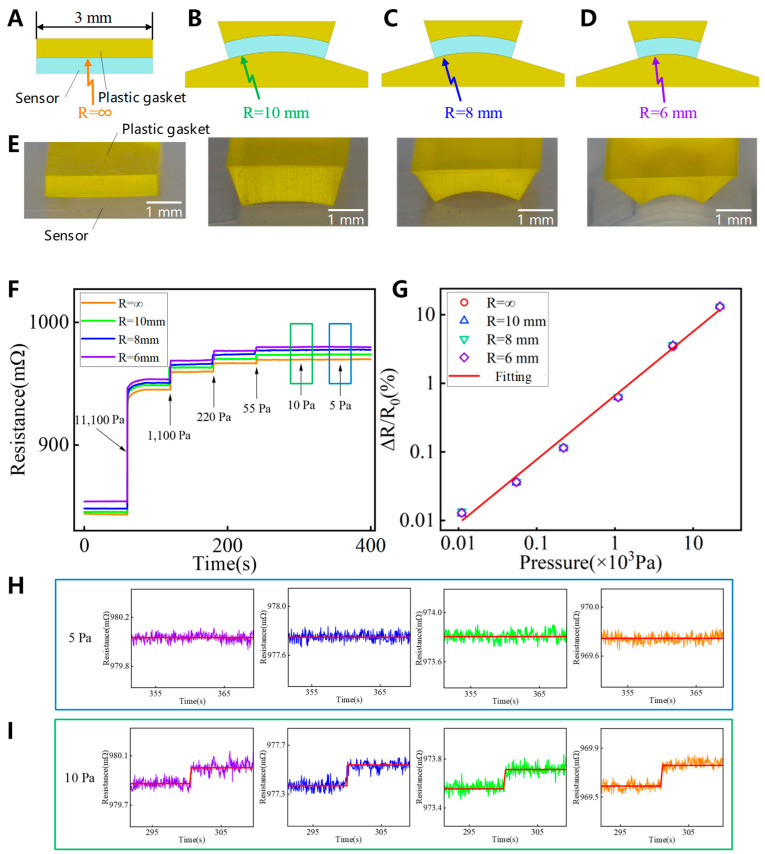
Bending test of array unit. Radius of curvature: (**A**) ∞, (**B**) 10 mm, (**C**) 8 mm, and (**D**) 6 mm. (**E**) Image of resin pad. (**F**) Pressure resolution on a curved surface. (**G**) Linear response of sensing unit on a curved surface. (**H**,**I**) Pressure response of different surfaces under 5 Pa and 10 Pa.

**Figure 6 sensors-25-00037-f006:**
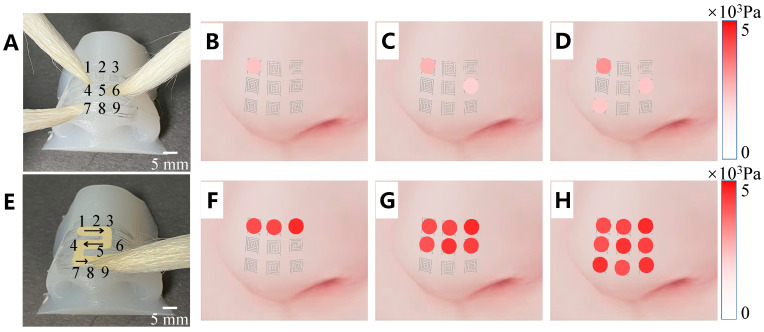
Performance test of a bionic model of the nasal tip. (**A**–**D**) Image of multitouch sensing on model and results. (**E**–**H**) Image of sliding sensing on model and results.

**Table 1 sensors-25-00037-t001:** Performance Comparison.

Work	1/Lower Detection Limit (1/Pa)	Sensing Order of Magnitude	1/Pressure Resolution (1/Pa)	1/Time Resolution (1/s)	Frequency (Hz)	Speed Perception (mm/s)	Minimum Working Curvature (mm)
This work	1/10	10^4^	1/10	1/0.05	1~50	8~54	6
Ref. [36]	1/100	10^4^	1/20	1/0.04	1~50	8~75	6
Ref. [51]	1/12	10^3^	1/12	1/0.13	0.25~2	-	-
Ref. [52]	1/25	10^4^	1/25	1/0.22	0.1~1	-	-
Ref. [53]	1/10	10^5^	1/10	1/0.04	-	-	-
Ref. [54]	1/100	10^5^	1/100	1/0.08	-	-	-

**Table 2 sensors-25-00037-t002:** Multitouch force for nose bionic model (×10^3^ Pa).

Position	B	C	D
1	1.20	1.52	2.14
6	-	0.94	1.13
7	-	-	1.10

**Table 3 sensors-25-00037-t003:** Sliding force table for upper arm bionic model (×10^3^ Pa).

Position	F	G	H
1	3.49	3.49	3.49
2	3.57	3.57	3.57
3	4.12	4.12	4.12
4	-	3.35	3.35
5	-	3.92	3.92
6	-	3.69	3.69
7	-	-	3.99

## Data Availability

The authors declare that the primary data supporting the findings of this study are available within the paper and its Appendix A. Additional data are available from the corresponding authors upon reasonable request.

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
