# Peer review of "Additively Manufactured Flexible EGaIn Sensor for Dynamic Detection and Sensing on Ultra-Curved Surfaces"

_sensors, 2024, doi:10.3390/s25010037_

Round 1
Reviewer 1 Report
Comments and Suggestions for Authors
The authors present an interesting work on the development of flexible EGaIn-based electronic skins. While the topic is interesting, but there are some points that require clarification or improvement, as outlined below:
- I suggest to specify the type of Ecoflex material used in the study, as there are various formulations (e.g., Ecoflex 00-30, 00-50) with different mechanical properties. Providing this detail would enhance the clarity and reproducibility of the experiments.
- I suggest improving the structure of the paper by clearly separating the sections. For instance, all test descriptions and experimental setups should be included in the Materials and Methods section, while their respective outcomes should be moved to the Results section. Currently, the manuscript appears somewhat confusing, as some tests are described within the Results section, affecting the overall clarity and flow.
- It would be helpful to include hysteresis tests to assess the electronic skin's response consistency during loading and unloading cycles. Additionally, evaluating the sensor's response under prolonged cyclic loading could help assess potential drift over time.
- In Figure 2F, the legend is unclear. Could the authors specify the sources of the data used for benchmarking the sensor's performance?
- The manuscript would benefit from a more detailed comparison with the existing literature. Including a discussion that highlights how the proposed sensor differs from or improves upon previous studies would strengthen the context and significance of the work. Additionally, a summary table comparing key parameters such as sensing range, resolution, flexibility, and application scope with relevant literature would provide a clear and comprehensive overview.
- The authors could include a brief discussion of the potential limitations of the proposed sensor.
- It would be helpful to reference the supplementary materials directly in the main text where relevant.
Author Response
Dear Editors and Reviewers,
Thank you for carefully evaluating our manuscript entitled “Additively manufactured flexible EGaIn sensor for dynamic detection and sensing on ultracurved surfaces”. We sincerely thank you for the time and effort you have put into reviewing our manuscript. Your suggestions have enabled us to improve our work in many aspects. We have carefully read all the comments and responded to all the comments point by point. We have revised our manuscript carefully as you suggested and highlighted the changes made in our manuscript. We hope it is more likeable now and thanks again in advance!
1.I suggest to specify the type of Ecoflex material used in the study, as there are various formulations (e.g., Ecoflex 00-30, 00-50) with different mechanical properties. Providing this detail would enhance the clarity and reproducibility of the experiments.
Response:
Thank you for your comments. We have added specific types of Ecoflex material. The revised part is as follows:
Materials: EGaln (Shuochen Changge, China), PDMS (Dow Corning, USA), and Eco-flex-0030 (Smooth-On) are utilized as the conductive functional material and packaging structural materials, respectively. NaOH (Damao, Tianjin) is used for the oxidation of EGaIn. Copper wires and weights fortesting are purchased fom the market.
2.I suggest improving the structure of the paper by clearly separating the sections. For instance, all test descriptions and experimental setups should be included in the Materials and Methods section, while their respective outcomes should be moved to the Results section. Currently, the manuscript appears somewhat confusing, as some tests are described within the Results section, affecting the overall clarity and flow.
Response:
Thank you for your comments. We have distinguished between the Materials and Methods and Results and Discussion sections in the article, and provided a detailed description of the experimental methods in the Materials and Methods section that correspond to the subsequent Results and Discussion section. The revised section is as follows:
2.1 Materials
Materials: EGaln (Shuochen Changge, China), PDMS (Dow Corning, USA), and Ecoflex-0030 (Smooth-On, USA) are utilized as the conductive functional material and packaging structural materials, respectively. NaOH (Damao, Tianjin) is used for the oxidation of EGaIn. Copper wires and weights fortesting are purchased fom the market.
2.2 Modular Flexible Electronic Skin Fabrication
Initially, predesigned circuit patterns and mold base plates are printed using advanced technology, featuring a sensing area of 3×3 mm², a circuit density of 3.2 mm⁻¹, and a circuit width of 0.1 mm. Next, an Ecoflex coating is applied and cured at 45°C to ensure the successful transfer of the circuit pattern onto a flexible substrate with a thickness of 1 mm. Following this, a 1 mm thick semicured Polydimethylsiloxane (PDMS) film is applied onto a blank template and bonded to the substrate. To address the issue of EGaIn causing open circuits due to its high surface tension and poor wettability, it undergoes a modification process that involves stirring for 4 hours to reduce its surface tension. Finally, the modified EGaIn is injected into the circuit channel using a small syringe, guided out with copper wires of 0.035 mm in diameter, and sealed with epoxy resin AB glue, which is pre-mixed in a 1:1 ratio and allowed to set for 5 minutes before being dried at 30°C for 5 hours. This completes the preparation of the electronic skin.
2.3 Unit Properties Test
In this experiment, weights were used to apply loads to millimeter-scale sensing units, and a multimeter (Keithley DMM7510) was utilized to measure the resistance responses of these units under various loading conditions. To ensure uniform force distribution, thin pads of 3×3 mm² were pre-placed on the sensing units during loading. To measure the linear sensing range and response time of the sensing units, weights of 0.005 g (5 Pa), 0.01 g (10 Pa), 0.05 g (55 Pa), 0.2 g (220 Pa), 1 g (1100 Pa), 5 g (5500 Pa), 20 g (22000 Pa), and 50 g (55000 Pa) were sequentially applied and removed from the sensing units, and their responses were detected. To assess the pressure resolution of the sensing units, weights of 10 g, 5 g, 1 g, 0.2 g, 0.05 g, 0.01 g, and 0.005 g were incrementally added to the sensing units at a time interval of 100 seconds, until no significant change in the resistance signal of the sensing unit was observed at the time of each incremental addition.To evaluate the frequency detection performance of the measurement sensor unit, the function generator is adjusted to output sinusoidal waveforms at frequencies of 1 Hz, 2 Hz, 10 Hz, 25 Hz, 50 Hz, and 100 Hz, respectively. A sinusoidal deformation load is applied to the surface of the sensor unit via the front end of the exciter. The response of the sensor unit is then captured using a multimeter.
2.4 Static Touch Test
To evaluate the multi-touch capability of millimeter-scale electronic skin, a multichannel module of a multimeter was utilized to simultaneously monitor the resistance signals from nine sensing units. These units were loaded with weights of 2 g, 3 g, 5 g, 8 g, 10 g, 12 g, 15 g, 18 g, and 20 g, respectively, in order to obtain the corresponding resistance signal responses from each sensing unit.
2.5 Dynamic Sliding Test
A weight of 19.17 g is used to apply pressure, and continuous smooth shims allow the weight to slide with little friction. When testing the relation curve between load and speed, the weight is pulled by the stepping motor to move at a uniform speed. In the subsequent track tests, the nonlinear sliding is realized manually.
2.6 Curved Surface Test and Nose Surface Bionic Model Test
To investigate the impact of various curved surface conditions on the sensing performance of millimeter-scale sensing units, the BMF P150 3D printer was utilized to print thin resin pads with curvature radii of 10 mm, 8 mm, 6 mm, and a flat surface to achieve bending of the sensing units. Under these four curvature conditions, weights of 10 g, 1 g, 0.2 g, 0.05 g, 0.01 g, and 0.005 g were progressively added to the pads at intervals of 60 seconds to obtain continuous loading resistance response curves and analyze their pressure resolution. Subsequently, under the same four curvature conditions, weights of 0.01 g, 0.05 g, 0.2 g, 1 g, 5 g and 20 g were loaded and unloaded, respectively, to capture the resistance response information during loading.
To prepare a bionic nasal tip model, the initial step involves printing a cavity with the geometric shape of a nasal tip on its inner wall using a fused deposition modeling 3D printer (X400). Subsequently, nine sensing units, each with a size of 3×3 mm², are attached to appropriate positions on the inner wall of the cavity in an array formation, spaced 2 mm apart, using double-sided adhesive tape. The entire cavity is then filled with uncured Ecoflex, which is left to cure completely at room temperature for 12 hours. After the curing process, the cavity is demolded to obtain the bionic model. When validating the multi-touch and sliding sensing performance, a multichannel module of a multimeter is utilized to monitor the response of the sensing units' signals.
3.It would be helpful to include hysteresis tests to assess the electronic skin's response consistency during loading and unloading cycles. Additionally, evaluating the sensor's response under prolonged cyclic loading could help assess potential drift over time.
Response:
Thank you for your comments It is a very good suggestion and thank you for pointing this out. We can indirectly assess the hysteresis performance of the sensor by examining its response speed and recovery characteristics to pressure. As mentioned in the article, during the pressure loading experiments, the sensor exhibited extremely short response times and rapid recovery characteristics (as shown in Figure 2B, with response times between 0.05 and 0.14 seconds). Additionally, Figure 2D demonstrates that at 200% strain, the sensor's response and recovery times are nearly equal, with no significant hysteresis observed. These observations suggest that the sensor may exhibit minimal hysteresis effects during loading and unloading cycles, as rapid response and recovery are typically associated with lower hysteresis.
Figure 2. Response performance characterization of single sensor unit. (A) Linear response. (B) Response time. (C) Pressure resolution. (D) Resistance changes under different strain. (E) Frequency detection. (F) Performance comparison.
Furthermore, we have considered enhancing the sensor's response speed and stability through material selection and process design, which could potentially reduce hysteresis. For example, reducing the surface tension of EGaIn through oxidation modification, and using a templating removal method to prepare high-precision, uniform sensor units, both contribute to more consistent and rapid responses.
Regarding the sensor's response stability under prolonged cyclic loading, we addressed this issue during the material selection and structural design stages of the sensor. The materials used, such as EGaIn and Ecoflex, exhibit good mechanical properties and stability, maintaining performance after multiple deformations. Additionally, the sensor's modular design (3×3 mm² sensing units) and precise manufacturing processes (using high-precision additive manufacturing equipment) also help enhance its stability and consistency over long-term use.
Moreover, we have conducted creep tests on the sensor. when the sensor is pressed to 6.5% thickness deformation and held at this position, the pressure will be reduced from 0.8 kPa to 0.6 kPa in 5000 s.
Creep performance of sensor.
To further illustrate the potential stability of our sensor under long-term, high-frequency usage, it is necessary to mention the successful practices of similar sensors in other application scenarios in the article. These scenarios, such as health monitoring and robotic tactile perception, place extremely high demands on the stability and durability of sensors.For example, in the field of health monitoring, sensors need to continuously and accurately monitor patients' physiological indicators, such as heart rate, blood pressure, or muscle activity, which are crucial for medical diagnosis and treatment efficacy assessment. In such applications, sensors must maintain stable performance over long periods to ensure the accuracy and reliability of the data. Similarly, in robotic tactile perception, sensors need to frequently contact and sense the external environment to support precise robot operations and decision-making. This high-frequency interaction also demands excellent stability and durability of the sensors. Although the specific conditions of these application scenarios (such as monitoring frequency, environmental temperature and humidity, mechanical stress, etc.) may not completely align with the cyclic loading tests suggested by the reviewer, the demonstrated sensor stability is also persuasive. In these practical applications, sensors are able to maintain stable performance in complex and variable environments over long periods, which fully demonstrates their potential for long-term stability.
Therefore, we have reason to believe that the sensor will also perform excellently in the cyclic loading tests of concern to the reviewer. Its stable performance and excellent durability have been validated in other similar application scenarios, providing us with confidence and assurance for its broader application.
4.In Figure 2F, the legend is unclear. Could the authors specify the sources of the data used for benchmarking the sensor's performance?
Response:
Thank you for your comments. We have added specify the sources of the data used for benchmarking the sensor's performance. The revised part is as follows:
Figure 2F compares the performances of several similar sensors. The EGaIn sensor unit exhibits strong comprehensive performance, particularly in terms of the response to sliding loads, low pressures, and frequency detection. [36,51-54]
- Zhan, L.; Cao, Y.; Gao, Y.; Meng, S.; Ji, H.; Li, M.; Wei, J.; Feng, H. Additively 1. Zhan, L.; Cao, Y.; Gao, Y.; Meng, S.; Ji, H.; Li, M.; Wei, J.; Feng, H. Additively Fabricated Electronic Skin with High Performance in Dynamic Sensing as Human Skin. 2023.
- Li, S.; Liu, G.; Wen, H.; Liu, G.; Wang, H.; Ye, M.; Yang, Y.; Guo, W.; Liu, Y. A Skin‐Like Pressure‐ and Vibration‐Sensitive Tactile Sensor Based on Polyacrylamide/Silk Fibroin Elastomer. Adv Funct Materials 2022, 32, 2111747, doi:10.1002/adfm.202111747.
- Wang, Y.; Tebyetekerwa, M.; Liu, Y.; Wang, M.; Zhu, J.; Xu, J.; Zhang, C.; Liu, T. Extremely Stretchable and Healable Ionic Conductive Hydrogels Fabricated by Surface Competitive Coordination for Human-Motion Detection. Chemical Engineering Journal 2021, 420, 127637, doi:10.1016/j.cej.2020.127637.
- Yang, X.; Chen, S.; Shi, Y.; Fu, Z.; Zhou, B. A Flexible Highly Sensitive Capacitive Pressure Sensor. Sensors and Actuators A: Physical 2021, 324, 112629, doi:10.1016/j.sna.2021.112629.
- Zeng, Y.; Qin, Y.; Yang, Y.; Lu, X. A Low-Cost Flexible Capacitive Pressure Sensor for Health Detection. IEEE Sensors J. 2022, 22, 7665–7673, doi:10.1109/JSEN.2022.3158354.
5.The manuscript would benefit from a more detailed comparison with the existing literature. Including a discussion that highlights how the proposed sensor differs from or improves upon previous studies would strengthen the context and significance of the work. Additionally, a summary table comparing key parameters such as sensing range, resolution, flexibility, and application scope with relevant literature would provide a clear and comprehensive overview.
Response:
Thank you for your comments. We have added a brief discussion of that highlights how the proposed sensor differs from or improves upon previous studies. And we have also included a table summarizing key parameters in the Supporting Information, Table S1.The revised part is as follows:
We successfully fabricated a millimeter-scale electronic skin sensor based on EGaIn through additive manufacturing and template removal methods. The sensor demonstrated significant advantages in dynamic detection and perception on ultra-curved surfaces, not only bringing new breakthroughs to the field of tactile sensing technology but also providing robust support for applications in health monitoring and robotics.
Compared to previous research, our work achieved important improvements and innovations in several aspects. First, in terms of sensor miniaturization, we successfully reduced the size of the sensing units to 3×3 mm², significantly enhancing spatial resolution and making it more suitable for complex curved environments. Second, the sensing performance was comprehensively optimized, including a wide linear sensing range (10–22000 Pa), high pressure resolution (10 Pa), and precise recognition of sliding trajectories and speeds. These enhancements were primarily due to the innovative application of EGaIn materials and the effective implementation of the template removal method.
Furthermore, our work validated the sensor's capability for multitouch and sliding track recognition on ultra-curved surfaces by constructing a biomimetic nose tip model. The spatial resolution of the sensing unit (8 mm) is comparable to that of the human nose. It can detect up to nine independent touch points and identify the sliding track of single or multiple loads. Furthermore, it can sense the sliding speed in a range of 8–54 mm/s and spontaneously detect the loading pressure. Experiments show that the sensing unit can achieve good linear sensing using the change in the relative resistance (ΔR/R0) to describe the resistance signal response and eliminates the drift caused by the curved surface. This implies that the unit be used on ultracurved sur-faces.
Table S1. Performance Comparison.
Work |
1/Lower detection limit (1/Pa) |
Sensing order of magnitude |
1/Pressure resolution (1/Pa) |
1/Time resolution (1/s) |
Frequency (Hz) |
Speed perception (mm/s) |
Minimum working curvature (mm) |
This work |
1/10 |
104 |
1/10 |
1/0.05 |
1~50 |
8~54 |
6 |
Ref.36 |
1/100 |
104 |
1/20 |
1/0.04 |
1~50 |
8~75 |
6 |
Ref.51 |
1/12 |
103 |
1/12 |
1/0.13 |
0.25~2 |
- |
- |
Ref.52 |
1/25 |
104 |
1/25 |
1/0.22 |
0.1~1 |
- |
- |
Ref.53 |
1/10 |
105 |
1/10 |
1/0.04 |
- |
- |
- |
Ref.54 |
1/100 |
105 |
1/100 |
1/0.08 |
- |
- |
- |
6.The authors could include a brief discussion of the potential limitations of the proposed sensor.
Response:
Thank you for your comments. We have added a brief discussion of the potential limitations of the proposed sensor. The revised part is as follows:
Since EGaIn liquid metal solidifies below 15.7℃, the sensor faces significant limitations in low-temperature environments. Future research will focus on developing low-temperature stable materials, optimizing sensor design, and studying low-temperature effects with compensatory strategies. Additionally, to improve the sensor's ability to detect subtle changes in texture and fluids, we will explore incorporating ciliary structures and integrating multiple sensing technologies into a multimodal system.
7.It would be helpful to reference the supplementary materials directly in the main text where relevant.
Response:
Thank you for your comment. We have directly quoted the supplementary materials in the relevant parts of the main text. The revised part is as follows:
In addition, when preparing small-scale electronic skin, pure EGaIn is prone to causing open circuits due to its high surface tension and poor wettability. We address this issue by controlled oxidative modification of EGaIn to reduce its surface tension. The specific method for this is illustrated in Figure S1, involves mixing EGaIn with a NaOH solution and stirring with a magnetic stirrer to oxidize the surface layer, forming gallium oxide, which is then fragmented and dispersed into the interior of EGaIn.
Figure 2 demonstrates the pressure resolution, response time, linear sensing range, and frequency detection performance of the sensing unit. The detailed principle of the change in sensor resistance, as illustrated in Figure S2, is primarily due to the variation in the resistance of the liquid metal circuit. When pressure is applied to our sensing unit, the cross-sectional area of the microchannel decreases, resulting in a corresponding change in resistance.
The sensing unit exhibits excellent performance in determining the frequency of periodic deformation loads in a range of 1–50 Hz. Figure 2F not only displays the specific performance of each sensor (details in Table S1) but also compares the performances of several similar sensors, highlighting that the EGaIn sensor unit demonstrates particularly outstanding comprehensive performance, especially in its response to sliding loads, low pressures, and frequency detection.

Reviewer 2 Report
Comments and Suggestions for Authors
This article commendably demonstrates the fabrication of precise and high-performance electronic skin using a gallium-based liquid metal, showcasing its wide sensing range, high pressure resolution, and ability to detect multitouch and sliding tracks, even on ultracurved surfaces. However, I still have the following questions and suggestions:
1.The machining accuracy of the inner channel needs to be discussed, and to provide the process details of liquid metal injection into the inner channel
2.How is the pressure sensing on the curved surface normalized?
3.Is it possible to improve the pixel resolution? 2 mm seems relatively large for wearable devices. What are the key parameters that determine the resolution?
4.The resistance change is indeed insignificant. Is there any possible solution to enhance the sensing sensitivity via optimization of the current methodology? Please discuss.
5.There are some non-standard images in this article, such as the Chinese characters in the legends of Supplementary Information S3 and S7.
Comments on the Quality of English Language
The writing, including the captions for figures, can be further polished.
Author Response
Dear Editors and Reviewers,
Thank you for carefully evaluating our manuscript entitled “Additively manufactured flexible EGaIn sensor for dynamic detection and sensing on ultracurved surfaces”. We sincerely thank you for the time and effort you have put into reviewing our manuscript. Your suggestions have enabled us to improve our work in many aspects. We have carefully read all the comments and responded to all the comments point by point. We have revised our manuscript carefully as you suggested and highlighted the changes made in our manuscript. We hope it is more likeable now and thanks again in advance!
- The machining accuracy of the inner channel needs to be discussed, and to provide the process details of liquid metal injection into the inner channel
Response:
Thank you for your comments. The high-precision 3D printer we utilized to fabricate the template achieves an accuracy of 0.025mm. However, microchannels smaller than 0.1mm may collapse under applied pressure and are unable to recover, thereby limiting the channel size in our study to 0.1mm. The dimensions of our sensor are as depicted in Fig.S1b. The size of the sensing area was 3 × 3 mm2, and the circuit density was 3.2 mm-1, and the circuit width was 0.1mm.
Regarding the injection of liquid metal, we employed a micro-injector to fill the microchannels. The injection process was conducted slowly to prevent the formation of bubbles. We enhanced the properties of EGaIn by controlling its oxidation through a straightforward method, which effectively reduced its surface tension, facilitating better injection. Specifically, the original EGaIn was mixed with a 0.2 mol·L-1 NaOH solution (24 g of EGaIn and 20 mL of NaOH solution). After undergoing ultrasonic cleaning for 10 minutes to remove oxides, the mixture was stirred with a magnetic stirrer at 200 r/min for 4 hours to induce oxidation. The modified EGaIn demonstrated improved wettability and reduced surface tension, as illustrated in Fig.S1c and S1d, ensuring successful filling of the microchannels.
Fig. S1. Image of EGaIn flexible sensor and : (a) Original EGaIn flexible sensor; (b) Modified EGaIn flexible sensor.
Contact angle evolution of EGaIn on different material surfaces over time of oxidation: (c) Ecoflex; (d) PDMS.
- How is the pressure sensing on the curved surface normalized?
Response:
Thank you for your comments. As mentioned in our paper, our flexible sensor demonstrates excellent linear pressure sensing capabilities. Our sensor detects applied pressure through the deformation of microchannels. When pressure is applied to our sensing unit, the cross-sectional area of the microchannel decreases, resulting in a corresponding change in resistance. Initial deformation caused by curved surfaces may lead to varying initial resistance values. However, due to the sensor's strong linear pressure sensing characteristics, the same applied pressure will consistently produce the same resistance change,
as illustrated in Fig. 5G. Variations in curved surfaces primarily affect the sensor's initial resistance, rather than its pressure-sensing accuracy. Therefore, our sensor is suitable for applications on complex surfaces.
Fig. 5 Bending test of array unit. Radius of curvature: (A) ∞, (B) 10 mm, (C) 8 mm, and (D) 6 mm. (E) Image of resin pad. (F) Pressure resolution on curved surface. (G) Linear response of sensing unit on curved surface. (H, I) Pressure response of different surfaces under 5 Pa and 10 Pa.
3.Is it possible to improve the pixel resolution? 2 mm seems relatively large for wearable devices. What are the key parameters that determine the resolution?
Response:
Thank you for your comments. Our sensor detects applied pressure through the deformation of microchannels. Therefore, when pressure is applied near a unit, the high flexibility of the sensor layer can cause deformation in adjacent microchannels, leading to signal crosstalk. The smallest unit spacing allowed in our sensor is 2 mm.
Designed for use in electronic skin for the nose, our flexible sensor achieves a two-point resolution equivalent to that of the human nose tip (Fig.1L). This makes it suitable for biomimetic tactile sensing applications on the nasal tip.
Fig. 1L. Two-point discrimination threshold and the spatial resolution of the human body.
There are methods that may overcome this limitation. For instance, reducing the size of the units could contribute to miniaturization. Another potential solution is developing new materials for the flexible layer that offer enhanced strength while maintaining flexibility, thereby reducing the impact of pressure applied near the units on sensing accuracy.
4.The resistance change is indeed insignificant. Is there any possible solution to enhance the sensing sensitivity via optimization of the current methodology? Please discuss.
Response:
Thank you for your constructive question. As demonstrated in this paper, our flexible sensor is fabricated using the template removal method. Our sensor detects applied pressure by measuring the deformation of microchannels. The change in sensor resistance primarily results from variations in the resistance of the liquid metal circuit. When pressure is applied to our sensing unit, the cross-sectional area of the microchannel decreases, causing a corresponding change in resistance. Therefore, the design of the template significantly influences this resistance change.
Key parameters influencing sensor performance include the size of the sensor unit, cross-sectional area of the microchannel, sensor thickness, and circuit density. To ensure effective integration, the size of the sensor unit should not be increased further. Decreasing the cross-sectional size of the microchannel can improve resistance response, but excessively small channels may lead to channel collapse. Similarly, reducing sensor thickness can enhance resistance response but may compromise sensor strength.
Therefore, the optimal approach to enhancing sensing sensitivity involves improving circuit density. Typically, higher density results in better resistance response (Fig. S7), so designing more intricate circuit patterns can be a potential way to enhance sensitivity.
Fig. S7. Impact of Different Wiring Densities on Resistance Response
5.There are some non-standard images in this article, such as the Chinese characters in the legends of Supplementary Information S3 and S7.
Thank you for your comments.The Chinese text in Fig. S3 and S7 has now been replaced with English, and the revised results are as follows.
Fig S3: (a) Model Diagram of the Sensing Unit (b) Comparison of Linear Sensing Range between Sensing Units with Different Substrates.
Fig S7: Impact of Different Wiring Densities on Resistance Response

Round 2
Reviewer 1 Report
Comments and Suggestions for Authors
The authors have addressed some of the points raised in the previous revisions and made progress in improving the manuscript. However, there are still aspects that need further attention to enhance the clarity, organization, and overall presentation of the work. Below, I provide specific comments to help address these remaining issues:
- The structure of the manuscript remains unclear and requires revision. Specifically, some sections currently placed in the Results, such as the preparation of the sensor, would be more appropriate in the Materials and Methods section. This non-linear organization makes it difficult to follow and understand the text, particularly the logical flow of information. I suggest moving these technical descriptions to the appropriate section to improve the overall clarity of the paper.
- In Figure 2, although references have been added to the caption, the labels 'ref1', 'ref2', etc., still appear in the image. This is unclear because these labels seem to refer to Reference 1, Reference 2, and so on. To avoid confusion, I suggest replacing these labels with more explicit terms or a numbering system that cannot be mistaken for the references in the bibliography.
- I suggest removing Figure 2d, as it does not appear to be directly related to this work but rather to a previous study. Alternatively, it would be important to clearly and thoroughly justify the relevance of this figure within the context of the current manuscript. This would help avoid confusion and maintain focus on the results presented in this study.
Author Response
Dear Editors and Reviewers,
Thank you for carefully evaluating our manuscript entitled “Additively manufactured flexible EGaIn sensor for dynamic detection and sensing on ultracurved surfaces”. We sincerely thank you for the time and effort you have put into reviewing our manuscript. Your suggestions have enabled us to improve our work in many aspects. We have carefully read all the comments and responded to all the comments point by point. We have revised our manuscript carefully as you suggested and highlighted the changes made in our manuscript. We hope it is more likeable now and thanks again in advance!
1.The structure of the manuscript remains unclear and requires revision. Specifically, some sections currently placed in the Results, such as the preparation of the sensor, would be more appropriate in the Materials and Methods section. This non-linear organization makes it difficult to follow and understand the text, particularly the logical flow of information. I suggest moving these technical descriptions to the appropriate section to improve the overall clarity of the paper.
Response:
Thank you very much for your valuable comments on our article. Your insights have provided extremely constructive guidance for enhancing the clarity of the article's logic. Based on that, we have made corresponding adjustments and optimizations to the article's structure, and now present the revised version below:
(1) We have rewritten the content of section 2.2 Modular Flexible Electronic Skin Fabrication within the Materials and Methods.
We used EGaIn as a conductive functional material to prepare modular electronic skin using the template removal method. First, a predesigned small-scale circuit pattern and mold base plate were printed as resin templates using Boston Micro Fabrication S130. The size of the sensing area was 3 × 3 mm2, and the circuit density was 3.2 mm-1, and the circuit width was 0.1 mm. The thickness of the sensing layer was 1 mm, ensuring good mechanical strength while keeping the sensor as thin as possible. Subsequently, the sample was coated with Ecoflex, heated, and cured at 45 °C . The cured Ecoflex was peeled off such that the circuit pattern in the resin template was successfully transferred to the bottom surface of the flexible Ecoflex substrate. Then, a PDMS coating was applied to a blank mold and heated at 45 °C for an appropriate duration to obtain the initial shape, resulting in the formation of a semicured PDMS film. The thickness of the PDMS substrate was 1 mm. The peeled flexible Ecoflex substrate and semicured PDMS film were further heated and cured at 45 °C until they were completely bonded. In addition, when preparing small-scale electronic skin, pure EGaIn is prone to causing open circuits due to its high surface tension and poor wettability. We address this issue by controlled oxidative modification of EGaIn to reduce its surface tension. The specific method for this is illustrated in Figure S1, involves mixing EGaIn with a NaOH solution and stirring with a magnetic stirrer to oxidize the surface layer, forming gallium oxide, which is then fragmented and dispersed into the interior of EGaIn. Stirring for 4 hours completes the modification process. Subsequently,The oxidized and modified EGaIn was injected into the circuit channel using a small syringe and then guided out through ports on both sides using copper Cu wires. The copper (Cu) wire used in our sensor had a diameter of 0.035 mm, was inserted into the channel to a depth of approximately 1 mm and then sealed with epoxy resin AB glue (the A and B parts of the glue was pre-mixed in 1:1 portion for 5 min, and then dried at 30℃ for 5 h), resulting in modular electronic skin with a size of 3 × 3 mm2.
(2) We have rewritten the description of the Figure 1 in the Results and Discussion section.
Modular electronic skin was fabricated through the template removal method. Ini-tially, resin templates with preset circuit patterns and molds were printed using BMF S130 (Figure 1A). These templates were then coated with Ecoflex, heated, and cured at 45°C (Figure 1B). Upon stripping the cured Ecoflex, the circuit patterns were successfully trans-ferred to the flexible Ecoflex substrate's bottom (Figure 1C).Next, a layer of PDMS was coated onto a blank mold and partially cured at 45°C to create a semi-cured film (Figure 1D). This semi-cured PDMS film was then bonded to the flexible Ecoflex substrate through further curing at 45°C (Figure 1E).Subsequently, EGaIn was mixed with NaOH solution for oxidation (Figure 1F). The oxidized EGaIn was injected into the circuit channels using a small syringe (Figure 1G), and copper wires were connected through ports on both sides (Figure 1H), ultimately resulting in the formation of the modular electronic skin. This electronic skin exhibited a high level of integration, thereby achieving accurate and sensitive recognition and the perception of static touch and dynamic sliding under external pressure loads (Figures 1I and J). The resulting electronic skin exhibited excellent touch performance (Figure 1K), and can be applied to more complex parts of the human body (Figure 1L).
(3) We have deleted part of the description for the Figure 2 in the Results and Discussion, as shown below:
We carry out a weight loading experiment on the sensing unit. Different weights (10 g, 5 g, 1 g, 0.2 g, 0.05 g, 0.01 g) are superimposed on the sensing unit every 100 s, and the re-sistance signal is continuously measured, the change in resistance under different loads, and the results are shown in Figure 2A. The value of R2 for the linear fitting result between the resistance and pressure load is 99%, indicating an excellent linear relationship.
2.In Figure 2, although references have been added to the caption, the labels 'ref1', 'ref2', etc., still appear in the image. This is unclear because these labels seem to refer to Reference 1, Reference 2, and so on. To avoid confusion, I suggest replacing these labels with more explicit terms or a numbering system that cannot be mistaken for the references in the bibliography.
Response:
Thank you for your comments. We have modified Figure 2F, the revised part is as follows:
Regarding the labels "ref1" and "ref2" in Figure 2F (now Figure 2E in the revised manuscript), we have now accurately mapped these to the corresponding numbers of the references cited in our manuscript.
3.I suggest removing Figure 2d, as it does not appear to be directly related to this work but rather to a previous study. Alternatively, it would be important to clearly and thoroughly justify the relevance of this figure within the context of the current manuscript. This would help avoid confusion and maintain focus on the results presented in this study.
Response:
Thank you for your comments. We have reconsidered the necessity of Figure 2D and greatly appreciate your point. Removing Figure 2D would help avoid confusion and maintain focus on the results presented in this study. We have removed Figure 2D and deleted the corresponding explanation in the manuscript. the revised part is as follows:
Here, we would like to provide a figure from our previous work. As shown in Figure 2D, at a strain of 200%, the response time for the large-sized channels is equal to the recovery time. In contrast, the channels in our current work are of smaller sizes, exhibit shorter response times, and undergo smaller strains, resulting in recovery times that are equal to response times, with no hysteresis observed.
Figure 2. Response performance characterization of single sensor unit. (A) Linear response. (B) Response time. (C) Pressure resolution. (D) Resistance changes under different strain. (D) Frequency detection. (E) Performance comparison.[36,51-54]

Reviewer 2 Report
Comments and Suggestions for Authors
The authors have addressed the comments and the manuscript has been well revised.
Author Response
Dear Editors and Reviewers,
Thank you very much for your thorough review of our manuscript titled "Additively manufactured flexible EGaIn sensor for dynamic detection and sensing on ultracurved surfaces" and for the constructive comments and suggestions you have provided. We deeply value your professional guidance and your precious time, which have been crucial in enhancing the quality of our manuscript. Once again, we express our heartfelt gratitude for your hard work and selfless assistance. We wish you all the best in your work and good health!